# An exploratory study on the impact of ESG on business performance—Focusing on listed companies in Korea and Taiwan

**Shaojie Fan**[ORCID]*

Gachon University, Seongnam, South Korea

* 202255056@gachon.com

**Data Availability Statement:** All relevant data are within the paper and its Supporting Information files.

## Abstract

In the context of the ESG era, this study provides an in-depth analysis of the ESG practices of listed companies and their impact on business performance in Korea and Taiwan, two of the Four Little Dragons economies in Asia. Although these two regions are similar in terms of economic size, they show significant differences in their ESG implementation strategies and effects. Based on the Bloomberg database, this study empirically analyzes data from 113 Taiwanese and 113 Korean firms, using Tobin's q ratio as a measure of business performance. The findings show that there is complexity in the association between ESG scores and firms' business performance. In South Korea, government policies and large conglomerates contribute significantly to ESG practices, while in Taiwan, the economic structure dominated by SMEs has led to different characteristics of ESG practices. All of these differences reflect the influence of intra-firm factors on performance. The findings of this study not only enrich the theoretical foundation of the relationship between ESG and business performance, but the findings provide valuable regional insights and recommendations for international investors, corporate managers, and policymakers in the Asia-Pacific region to implement ESG strategies, especially when considering the specific market environment, economic structure, and internal factors of the firms they operate in order to achieve sustainable growth and competitive advantage.

## 1. Introduction

### 1.1 The question is posed

In the context of current global economic and social development, environmental, social, and governance (ESG) practices have become a key measure of corporate responsibility and sustainability (Drempetic S, Klein C&Zwergel B, 2020) [1]. Given this, this study focuses on an in-depth examination of the performance of listed companies in two Asian economies, South Korea and Taiwan, in terms of their ESG practices and their specific impact on corporate business performance. Considering the impact of global ESG standards on firms, this study aims to analyze and compare how Korean and Taiwanese firms implement ESG responsibilities and

**Funding:** The author(s) received no specific funding for this work.

**Competing interests:** The authors have declared that no competing interests exist.

the impact of these practices on long-term sustainability and social responsibility of firms, therefore, the issues examined in this paper are specified as follows.

First, this study focuses on the dimension of the relationship between ESG practices and firms' business performance by analyzing the performance of listed firms in South Korea and Taiwan in terms of ESG practices through empirical research, which will pay special attention to how these practices affect firms' business performance. By examining this dimension, this paper aims to reveal ESG practices through empirical research to reveal the specific impacts of ESG practices on firms' business performance in different social contexts to deepen the in-depth understanding of the effective implementation of ESG in different environments.

The second is the difference between South Korea and Taiwan in terms of the specific implementation of ESG. This study will aim to explore the differences in ESG implementation between Korea and Taiwan and how these differences affect firms' business performance. This includes the consideration of various aspects such as firm size, growth dimensions, and firm assets as a way to explore how these factors shape unique ESG practice patterns in the two different economies.

In summary, this study aims to provide corporate managers, policymakers, and academics with insights into the complex relationship between ESG practices and business performance, and to provide new perspectives and recommendations for future research and practice in this area. After considering the global economic and social changes, especially especially after the COVID-19, it is necessary to provide insights into exploring how the role and importance of ESG have changed and how Korean and Taiwanese firms have adapted to these changes is also a key direction of this study. In addition, through the comparative analysis of ESG practices at each element level in two different economies, Korea and Taiwan, this study will provide recommendations and references for companies to effectively integrate and implement ESG strategies in different environments, and it is also expected that this study will provide valuable references for companies in the Asia-Pacific region to achieve effective operational results in terms of sustainability and social responsibility practices.

## 1.2 Background and necessity of the study

In the current context of globalized economic and social development, the challenges faced by corporations are not only limited to traditional economic competition, but also extend to environmental, social, and governance (ESG) responsibilities, and ESG standards have become a key indicator of corporate sustainability and responsibility, demonstrating the commitment of corporations to environmental protection, social responsibility, and good governance, as well as providing investors, consumers, and other stakeholders with an opportunity to assess the long-term value and performance of corporations. stakeholders with an important basis for assessing the long-term value and risk of a company (Dorfleitner G, Halbritter G & Nguyen, 2015) [2]. With the increasing global focus on environmental protection, social justice and governance transparency, ESG practices have become an integral part of corporate operations.

In the Asian economic landscape, South Korea and Taiwan, as important economic entities, have attracted much attention for their performance in ESG practices. Although the two places are similar in terms of economic scale, they show significant differences in the specific implementation and effectiveness of ESG practices. This difference may stem from the differences in economic structure, policy environment, and cultural background between the two places. The Korean economy is dominated by large conglomerates (zaibatsu), while Taiwan is dominated by small and medium-sized enterprises (SMEs), and this structural difference has led to the two regions adopting different strategies and responses when facing ESG challenges and opportunities (Yoon, Lee & Cho, 2001) [3]. In addition, South Korea and Taiwan also share

certain similarities, such as similar governance structures and comparable international influence of firms in both countries (Chen & Yang, 2020) [4]. Although there are differences in the corporate governance structures and models of the two countries, they are both facing issues such as improving corporate transparency and strengthening the protection of shareholders' rights and interests; both Taiwan and South Korea have a certain status and influence in the international community, and their ESG development levels and policy practices are exemplary and useful for other countries and regions. These commonalities provide valuable research perspectives for studying the role of corporate governance in ESG development.

ESG is growing in importance globally in the post-epidemic era, and corporate behavior and performance in environmental, social, and governance aspects have received unprecedented attention from the international community. This trend has led to the need for corporations to not only take on more social responsibility within their own countries in the future but also to demonstrate their alignment with the global sustainable development goals in the international arena (Abdul & Alsayegh, 2021) [5]. Given this, an in-depth study of the differences in ESG practices among the different economies in the Asia-Pacific region, especially South Korea and Taiwan, is particularly important to understand how they have adapted to adaptive issues amidst the global environmental and political-economic changes, and have gained experiences and lessons learned in terms of sustainable development.

Therefore, by comparatively analyzing ESG practices in South Korea and Taiwan, this study aims to reveal how these differences affect firms' business performance and provide an in-depth understanding of the factors specific to these regions. In addition, ESG practices of Taiwan's unique economic structure, especially its SME-oriented economic development model, such research will provide an important reference for evaluating and guiding the practices of SMEs in other countries and regions in the Asian region, and it will also provide international business managers, policymakers, and academics with insights into the relationship between ESG practices and business performance in the Asia-Pacific region, the promotion of Asia-Pacific It can also provide international business managers, policymakers, and academics with new perspectives and development proposals on the relationship between ESG practices and business performance, and the promotion of effective business operations in the Asia-Pacific region in terms of sustainability and social responsibility practices.

## 1.3 Purpose and significance of the study

The study will focus on analyzing the impact of ESG practices of listed companies in South Korea and Taiwan on key factors including firm size, growth rate, and shareholders' equity, as well as how these factors are reflected and managed in the day-to-day operations of the firms. By exploring the specifics of these practices in depth, the study aims to reveal how they affect firms' overall business performance, thereby providing firms with more effective business strategies and decision support. In addition, this paper compares the differences in the implementation of ESG responsibilities between South Korea and Taiwan, explores how these differences affect long-term corporate sustainability and social responsibility and analyzes the specific impacts of these differences on firms' business performance.

Taken together, this study hopes to provide corporate managers, policymakers, and academics with insights into the complex relationship between ESG practices and business performance. In particular, in the context of globalization, the case studies of South Korea and Taiwan provide references for corporations on how to better fulfill their social responsibilities in the international arena and align with the global SDGs to promote effective corporate operations in terms of sustainable development and socially responsible practices. Through in-depth analysis, it provides corresponding empirical references and developmental suggestions

on how companies can effectively integrate and implement ESG strategies in the face of environmental changes.

This study differs from prior studies in the following ways: first, this study is based on the macroeconomic context of the East Asian region, and creatively takes Taiwan and South Korea, which have comparable economic volumes, as the research subjects to explore the relationship between their operational performance and ESG. Second, existing studies have focused on the impact of corporate performance from the dimension of social responsibility, but they have often neglected to examine the association between ESG and corporate performance from a broader macro perspective, therefore, this paper explores the relationship between ESG and performance from a macro perspective by using the average score of each ESG dimension as the dependent variable.

## 2. Literature review

### 2.1 Definition, evolution, and theoretical framework of ESG

In the contemporary business environment, environmental, social, and governance (ESG), as an important framework for assessing corporate sustainability, has become the centerpiece of corporate strategic planning and external assessment. Such a framework not only focuses on the economic performance of a company, but also covers environmental protection, social responsibility, and a transparent and fair governance structure. To gain a deeper understanding of the theoretical foundations and practical dynamics of ESG, this paper will explore the development trends and key variables in this field from its definition, evolution, and current theoretical framework.

As early as 2004, the UN Global Compact first proposed the concept of ESG in Who Cares Wins, evaluated the operation of enterprises in this way, and proposed that the evaluation of enterprises through ESG as a scientific way of evaluation, and an important standard for the future evaluation of enterprises [6]. Lindgreen (2010) Discusses the role of CSR in the development of the company, and explains the motivation of corporate social responsibility, he believes that enterprises in the development process on the one hand, need to directly face consumers in the market, on the other hand, will also be subjected to the pressure from society and government policy, so enterprises should take the initiative to undertake social responsibility to shape the corporate image in the minds of consumers and reduce the pressure from the government and society [7]. The research on CSR continued until 2015 until the United Nations released the Sustainable Development Goals and ESG (Environment, Social, Governance) index 2015, the ESG evaluation system has specific indicators, as shown in Table 1 below:

ESG was initially proposed as a comprehensive assessment framework focusing on how companies can pursue economic efficiency while balancing environmental protection and social responsibility. With the deepening of the understanding of sustainable development, the concept and practice of ESG has gradually evolved and is now extended to a wider range of dimensions including corporate governance. In addition, for example, Liang H & Renneboog

Table 1. Factors representing the three dimensions of ESG.

| Environmental Protection | Social Responsibility | Corporate Governance |
| --- | --- | --- |
| Climate Change | Labor Compensation Standards | Board Diversity |
| Carbon Emissions | Workplace Diversity | Executive Compensation |
| Water Pollution | Occupational Health and Safety | Risk Control and Management |
| Biodiversity | Community Impact | Political Contributions |
| Industrial Waste Management | Labor Cost Burden | Bribery and Corruption |

(2017) emphasized the importance of ESG assessment from a global perspective, pointing out that ESG is a multidimensional issue covering the three pillars of environment, society, and governance [8]. Therefore, in the development process of ESG, while combing through the current literature currently supports ESG mainly the following three major theories.

First, sustainable development theory provides important theoretical support. It emphasizes that while achieving economic growth, companies should actively protect environmental resources and pay attention to social well-being to achieve long-term sustainability (Lokuwaduge & Heenetigala, 2017) [9].

Secondly, the economic externality theory takes a micro perspective and emphasizes the social costs and benefits arising from business activities, especially the impact on the environment. This theory states that market prices often do not fully reflect all social costs and benefits, so firms should consider these externalities in their business decisions (Consolandi, Phadke, Hawley,et al, 2020) [10].

Third, with the development of CSR theory, the social dimension of ESG has been further expanded. CSR theory emphasizes that while pursuing profits, firms should also take responsibility for their employees, consumers, communities, and the environment (Halbritter, Dorfleitner, 2015) [11].

In this framework, ESG scores become a key indicator of a company's overall performance on these three dimensions, providing investors and stakeholders with a way to quantify corporate sustainability. In terms of operational performance, such as Tobin's q-value and return on equity (ROE), are used to assess the impact of ESG practices on firms' financial performance, and prior research has shown how good ESG performance tends to correlate with higher operational performance.

Overall, ESG as a comprehensive assessment framework has been gradually developed and matured in theory and practice. It initially focused on balancing a company's economic performance with its environmental and social responsibilities, but over time, ESG has begun to focus more on the integration of environmental, social, and governance dimensions, and to remain dynamically adaptive to changes in the external environment. Today, ESG has become a key tool for measuring corporate sustainability. In addition, the analysis of key variables in ESG will include the ESG scoring system, business performance indicators, and market policy environment, which provides a solid foundation for corporate managers, policymakers, and academic researchers to deeply understand and apply ESG principles. With the growing global awareness of environmental protection and social responsibility, the importance of ESG is expected to rise and is becoming a key driver for companies to achieve sustainable development.

## 2.2 Comparison of ESG practices and experiences in the Asia-Pacific region

In environmental, social, and governance (ESG) practices in the Asia-Pacific region, South Korea and Taiwan show different characteristics and practice effects. The Korean government and large corporations, play an important role in promoting ESG practices, especially the role of chaebol companies in integrating ESG transparency should not be ignored. Yoon, Lee & Byun (2018) argue that the Korean government has emphasized the importance of ESG transparency through the development of the KESG Guidelines and has taken a series of measures. For example, the Korean government has taken a series of measures to promote the improvement of corporate governance structure and social responsibility practices to activate public transparency of ESG management and enhance the capabilities of SMEs in ESG [12]. South Korea's ESG practices have played an important role in improving corporate governance structure and promoting social responsibility, as well as facilitating ESG investment and information platforms.

In contrast, although Taiwan has gradually increased its influence in the global manufacturing industry and emphasized environmental protection and social responsibility, there is a gap between its overall ESG practices in terms of proactivity and systematization compared with those of South Korea. Taiwan's ESG practices focus more on the integration of local culture and community, and compared to South Korea's systematic and internationalized strategy, Taiwan's companies are more focused on complying with government decrees or investing in ESG sustainable management in a response-oriented manner. Taiwanese companies have positively impacted business performance when implementing CSR activities. However, this approach may lead to higher costs and less efficient practices (Lin & Hsu, 2023) [13]. ESG practices in South Korea and Taiwan not only emphasize the importance of environmental protection, social responsibility, and governance transparency but also demonstrate flexibility to adapt to local economic and cultural contexts (Kim, Shin, Lee, et al, 2024) [14]. Korea's proactive and systematic approach to ESG is reflected in the active promotion by the government and the full participation of large corporations, especially in the development and implementation of the K-ESG guidelines and the activeness of the ESG valuation and international consulting market, where Korea demonstrates the characteristics of its more open and diversified international market strategy.

Despite its proactivity, Korea's ESG practices have limitations. Although the government and large corporations have demonstrated proactivity in promoting ESG, this model centered on large corporations may lead to insufficient participation of SMEs in ESG practices (Yoon, Lee & Cho, 2021) [15]. In addition, ESG practices are overly reliant on government guidance and promotion by large corporations and lack market autonomy and innovation drive. In contrast, although Taiwan has slightly less systematic and internationalized practices, it pays more attention to the integration of local culture and community, which is conducive to the enhancement of corporate social responsibility and local market adaptability.

In summary, South Korea and Taiwan have their strengths and weaknesses in ESG practices. South Korea's initiative, systematic approach, and openness to international markets have brought significant results to its ESG practices, while Taiwan has performed better in terms of local culture and community integration. The experiences of the two places provide valuable lessons for other countries in the Asia-Pacific region in formulating and implementing ESG strategies and also allow for an analysis of how to effectively balance local characteristics and international standards in the context of globalization, which is an important developmental goal for future companies to achieve long-term sustainable development. Government policies and market environments in different countries have a significant impact on ESG practices, which in turn have an impact on the long-term success of companies. By analyzing these factors in depth, we can better understand how ESG practices have evolved in different economies and how they affect business performance.

## 2.3 Hypothesis formulation

**2.3.1 The relationship between corporate business performance and ESG.** In today's global corporate management field, the importance of environmental, social, and governance (ESG) practices has gradually come to the forefront as a key factor in assessing corporate performance. Numerous studies by scholars have focused on the impact of ESG practices on corporate business performance, revealing the positive and negative effects of this practice in different contexts as well as its trends and differences.

The first is that ESG practices have a significant positive effect on corporate business performance. Firms' CSR activities are positively correlated with R&D investment and sales and also show positive associations with corporate value indicators such as Tobin's Q (Cho Y, Kim S,

You, 2021) [16]. Zhong & Li's (2021) study further emphasizes that CSR activities help to enhance the firm's image and external reputation thereby directly contributing to business performance [17]. This view is supported by a study by Velte (2017), which found that ESG practices significantly improve the overall performance of German firms [18]. These studies suggest that ESG practices are not only related to the immediate financial performance of firms but also strongly associated with the long-term value of firms. However, ESG practices may also have some negative impacts.

Hwang, Kim & Jung's (2021) study also revealed a negative correlation between CSR activities and certain profitability indicators such as ROA and ROE, reflecting the fact that ESG practices may lead to higher costs in the short term [19]. In addition, Sahut (2015) year study pointed out that investments in ESG areas may adversely affect the short-term profits of firms [20]. In addition, regional differences in ESG practices were observed in this study. For example, Chung's study focused on firms in Taiwan, while Velte's study was based on data from German firms, suggesting that there are significant differences in the impact of different geographic locations, cultures, and economic backgrounds on ESG practices.

To summarize, although the academic community is divided on how and to what extent ESG affects business performance, the majority of the academic community agrees that ESG scores have a positive impact on business performance. Although the fulfillment of corporate social responsibility is bound to have costs, when the investment achieves certain results, it will benefit the enterprise through energy saving and emission reduction, obtaining a good reputation, attracting more investors, enhancing employee loyalty, obtaining policy support, and improving work efficiency in tangible or intangible ways, resulting in a positive benefit to compensate for the costs, realizing the scale effect of the investment, and then enhancing the enterprise's business performance, so this paper proposes The following hypotheses:

Hypothesis 1: ESG scores will have a positive impact on enterprise business performance

### 2.3.2 The relationship between enterprise business performance and other variables.

ESG practices, especially activities in the field of corporate social responsibility (CSR) and R&D investment, have a positive impact on corporate operating performance. These practices not only show a positive correlation with the firm's market value indicators, such as Tobin's Q, but also directly enhance the firm's operating results by enhancing the firm's brand image and external reputation, i.e., the return on shareholders' equity, as an indicator of the firm's operating performance, is positively enhanced by positive ESG practices (Naimy, Khoury & Iskandar, 2021) [21].

Firms of different sizes and structures also face different ESG challenges and opportunities. Senadheera&Withana (2021) noted that plutocratic firms face special internal and external challenges when implementing ESG [22]. The ESG framework covers several factors that affect a firm's long-term financial performance and sustainability, and therefore a firm's ESG characteristics will be a reflection of its ability to perform well over a longer time horizon to reflect the long-term value it can create in the future. For example, companies can increase revenues by developing specific markets and pursuing product differentiation strategies, reduce costs by reducing the risk of conflicts with external stakeholders and lowering the cost of raw materials, energy, and services, and fulfill their social responsibilities while achieving profitability, ultimately realizing the dual goals of economic benefits and environmental stewardship for the sustainable development of society.

In addition, ESG practices may have very different impacts on firms' short-term and long-term performance. Zhong & Li's (2021) study further emphasized that socially responsible activities can help to enhance firms' long-term sustainability and social responsibility [17]. In

the short term, ESG may lead to higher costs and lower profitability, while in the long term, it may enhance firms' market competitiveness and financial stability. Ahsan (2012) argued that to maintain and improve ROE, firms may need to engage in continuous research and development (R&D) and innovations to maintain a competitive advantage and meet market demands [23]. In summary, ROE has a profound impact on business performance, not only on the market value, investor confidence, and stock price performance of a company but also on its internal management and strategic decisions. Therefore, corporate management usually closely monitors ROE and uses it as an important reference indicator for corporate strategic planning and decision-making.

Therefore, the impact of ESG practices on corporate business performance is multidimensional, including both positive and negative effects. These effects vary by region, industry, enterprise size, and structure, and manifest themselves differently in different time scales, and the indicators to quantify these influences are mostly the return on shareholders' equity, enterprise size, enterprise growth rate, return on total assets, cost of debt capital cost of equity capital, liquidity ratio, total cash to assets ratio, income to assets ratio, ratio of independent directors, net operating profit, institutional shareholding ratio, research expenses, advertising costs, companies need to consider these complex factors when developing and implementing ESG strategies and make appropriate adjustments according to their circumstances.

Based on this, this paper proposes the following hypotheses:

Hypothesis 2: The return on equity has a positive impact on the business performance of enterprises

Hypothesis 3: Firm size positively affects business performance

Hypothesis 4: Enterprise growth rate has a positive impact on enterprise business performance

When firms make external investment decisions, their research expenses determine the quality and technological nature of the firm's products in the future period, advertising investment and research expenses have similarity, with the increase of advertising investment, the market's familiarity with the firm has increased, so the firms are more likely to obtain external financing, and the performance of the firms will be improved. Lee, Raschke & Krishen (2022) argued that With the application of ESG, market competition intensifies, and then advertising investment has a "crowding out" effect on R&D investment. Increasing investment in advertising reduces the incentives for R&D, the price of the enterprise's products, and the market share, but increases the incentives for R&D, the price of the products, and the market share of the other enterprises; when the intensity of market competition is low, the advertising investment has a "crowding out" effect on the R&D investment. When the intensity of competition in the market is low, advertising investment in R & D investment to produce a "crowd-in" effect, increases advertising investment, which will increase the pharmaceutical company's R&D incentives [24]. Therefore, this paper proposes the hypothesis:

Hypothesis 5: The proportion of research expenses and advertising expenses in the external investment decision of the enterprise has a significant positive impact on the enterprise's business performance.

Hypothesis 5–1: A firm's research expense share has a more significant effect on its business performance compared to its advertising expense share.

Prabowo & Simpson (2011) argued that independent directors have a certain degree of independence and can limit certain high-risk decisions of professional managers from the management's point of view, reducing the business risk of the enterprise, and thus enhancing

the performance of the enterprise [25]. In addition, institutional investors have a more complete organizational structure, usually have more resources and expertise, and can more effectively influence the strategic direction and operational efficiency of the company, and the existence of institutional investors increases the adaptability and market sensitivity of the enterprise to the external environment, thus improving the business performance of the enterprise to a certain extent. Therefore this paper argues:

Hypothesis 6: The corporate governance structure of enterprises, especially the proportion of independent directors and the proportion of institutional shareholding, has a positive impact on the operational performance of enterprises.

Hypothesis 6–1: The role of the proportion of institutional shareholding is more critical relative to the proportion of independent directors in enhancing the business performance of enterprises.

In summary, current research on ESG practices and corporate business performance suggests a multidimensional complexity in this area of inquiry. While a large number of studies have focused on assessing the impact of ESG on corporate performance in terms of social responsibility dimensions, they have often neglected to scrutinize the association between ESG and corporate performance from a broader macro perspective. In addition, existing studies mainly focus on specific regions or countries, lacking a comprehensive consideration of the impact of ESG practices in different geographical and cultural contexts. Therefore, taking a global perspective as a starting point, this paper provides an in-depth analysis of the relationship between ESG and corporate performance with the help of multivariate data sources, such as Bloomberg, to reveal the variability of ESG practices and their comprehensive impact on performance in different regions, sizes, and types of firms.

## 3. Research methodology

This paper focuses on the main factors affecting the business performance of enterprises and the mechanism of influence, and its main influencing factors this paper selected the ESG data disclosed by Bloomberg as the main research variables. This chapter divides the research methodology into three sections, the first section is the research sample selection and source of information, the second section is the model construction, and the third section is the definition of research variables.

### 3.1 Research sample selection and source of information

This study examines the business performance of enterprises and takes 113 enterprises in Taiwan and Korea of China as an example for this paper. Since Korea Corporate Governance Service incorporates too many local Korean factors when considering ESG activities, which is slightly different from the way ESG weights are calculated in Taiwan, the Bloomberg database is selected as the data source for this paper. The data source of this paper is the public ESG index scores of 113 firms disclosed in the Bloomberg database, of which 113 firms are Chinese-Taiwanese firms and 113 are Korean firms, and the above data are subsequently processed as follows:

First, due to the specificity of the business model, the companies in the financial industry have a certain gap with the non-financial industry companies, so the companies in the financial industry in the sample are excluded. Second, for the credibility of the research results of this paper, companies with listing status of ST and *ST are excluded. Third, to avoid the extreme data gap of individual companies is too large for the results to have an impact.

## 3.2 Modeling

The purpose of this paper is to understand the impact of ESG on the operational performance of Taiwanese and Korean companies, so the data of 113 Taiwanese and Korean companies from 2017–2022 are selected for this paper. The model used in this paper is the Panel Data Model, using measures including the Fixed Effect Model or Random Effect Model, and using Hausman Test to decide whether to use the FE or RE model.

The model set up in this study is shown below:

$$Y_{it} = \alpha_{it} + \alpha_1 X_{1,it} + \ldots + \alpha_1 X_{n,it} + \varepsilon_{it} \tag{1}$$

Where, i = 1,2. . . .113, represents the serial number of the research sample; t = 1,2,3,4,5,6 represents the time range of the research sample; is the operational performance of the ith company at time t, is the error term, and is the impact coefficient.

The parameter estimation of the regression model is more complicated than that of the linear model, which is calculated by adopting pooling to estimate the parameters. To deal with the heterogeneity of the samples of different firms, there are different ways to set the intercept term of the model, thus generating two different panel models:

**3.2.1 Fixed effects model.** The fixed effects model, also known as the Least Square Dummy Variable Model (LSDV), shows individual variability in the intercept term due to the variability between samples and between the same time series. In modeling, the intercept is usually assumed to be constant, and dummy variables are added to account for the effect of unobserved variables on the model, which can reduce the differences between individuals and show the differences to the intercept term using dummy variables. When there is a correlation between the residual term and the explanatory variables, the use of a random effects model will make the estimation results imprecise, and then a fixed effects model should be chosen. This is shown below:

$$Y_{it} = \sum_{j=1}^{N} \alpha_i D_{jt} + \sum_{K=1}^{K} \beta_K X_{K,it} + \varepsilon_{it} \tag{2}$$

$$Y_{it} = \alpha_i D_{jt} + \beta_K X_{K,it} + +\varepsilon_{it} \tag{3}$$

$$D_{jt} = \begin{cases} 1, & j = 1 \\ 0, & j \neq 1 \end{cases} \tag{4}$$

where is the business performance of the ith firm in cycle t, is the intercept term, is a dummy variable, and K is the K explanatory variable. If j = i, it is 1 and the rest is 0. If j≠i, it is = 0.

**3.2.2 Random effects model.** The random effects model is also called the Error Component Model (Error Component Model), the random effects model is used for samples in which cross-sectional and temporal data coexist, and its research focuses on the relationship between the whole, rather than the variability of the individual variables, the random effects model believes that the variability of the effects between the variables arises randomly, which is contrary to the fixed effects model, and therefore assumes that The coefficients are:

$$\alpha_i = \lambda + \mu_i \tag{5}$$

Where is a fixed unknown parameter, is the mean value of each variable affecting the outcome, and is an independent random variable, the regression model is:

$$Y_{it} = \lambda + \alpha_i D_{jt} + \beta_K X_{K,it} + \varepsilon_{it} \tag{6}$$

**3.2.3. Selection of model.** In this paper, when selecting the model, the Hausman test is used as a test, that is, the correlation between and is tested, which is also the biggest gap between the fixed effect model and the random effect model. If and are correlated, this leads to a bias in the estimates of the random effects model, which should be taken to the fixed effects model. If and are not correlated, the random effects model should be taken.

The test hypothesis is:

$$\begin{cases} H_0 : E(\mu_i, \ x_{kit}) = 0 \\ H_1 : E(\mu_i, \ x_{kit}) \neq 0 \end{cases} \tag{7}$$

$$H = (\beta_{RE} - \beta_{FE})'[\text{Var}(\beta_{RE}) - \text{Var}(\beta_{FE})](\beta_{RE} - \beta_{FE}) \sim x^2(K) \tag{8}$$

where is the coefficient estimate for the random effects model, is the coefficient estimate for the fixed effects model. is the explanatory variables are not related to the error term of the intercept, and is the explanatory variables are related to the error term of the intercept. When the statistical check value of Hausman test is greater than the chi-square check value under the degree of freedom as the number of explanatory variables, the fixed effect model should be adopted, and vice versa, the random effect model should be adopted.

H0: the intercept term is independent of the explanatory variables

H1: the intercept term is correlated with the explanatory variables

If H0 is accepted, then H0 is true and a random effects model should be adopted; if H0 is rejected, then H1 is true and a fixed effects model should be adopted.

## 3.3 Definition of research variables

**3.3.1 Explained variables.** Operational performance: Tobin's q ratio was first proposed by Brainard and Tobin (1968), which is defined as the ratio of the market value to the replacement cost of assets. Many scholars believe that Tobin's q has already taken into account the concepts of future cash flow and time value, so it is widely used as a method to measure the value of a company. Many scholars believe that Tobin's q has taken into account the concepts of future cash flow and time value, so it is widely used as a method to measure the value of a company, which is defined as the ratio of market value to the replacement cost of assets, and it is hoped that this indicator can be used to predict and plan the investment strategy of a company [26]. However, the original Tobin's q is too complicated to calculate, and in recent years, many scholars have proposed alternative formulas with certain explanatory power, therefore, this study uses the simple calculation of Tobin's q proposed by Surroca, Tribó, & Waddock (2010) to measure the performance of a firm's operations [27].

**3.3.2 Explanatory variables.** ESG Score: The independent variables of this study were adopted from the ESG database introduced by Bloomberg for 113 Chinese Taiwan companies and 113 South Korean companies in terms of environmental protection, social responsibility, and corporate governance, and the mean value of ESG was selected.

**3.3.3 Control variables.** Operating performance measures in Griffin & Mahon's (1997) study collated the financial performance variables used in the past empirical studies of CSR

and financial performance, of which the return on shareholders' equity, return on total assets, firm size, market excess compensation, market risk coefficient, and operating leverage are more scholars used in this study, which uses the return on shareholders' equity, firm size, This study uses return on equity, company size, debt ratio, research expenses, advertising expenses, proportion of independent directors and proportion of institutional shareholding as control variables and conducts statistical analysis [28].

1. *Return on shareholders' equity.* Griffin and Mahon believe that the return on assets is highly correlated with shareholders' equity and can reflect the operational efficiency of assets. Its calculation formula is:

$$ROE_{it+1} = \frac{NI_{it+1}}{Equity_{it+1}} * 100\% \tag{9}$$

Where t+1 represents the data of the following year, ROE is the return on equity, NI is net income after tax, and Equity is the average total shareholders' equity.

2. *Firm size.* Dutta, Narasimhan, and Rajiv (1999) pointed out that the size of the firm and the proportion of resources owned by the firm are positively correlated, and it is usually easier for large-scale firms to utilize limited resources to achieve economies of scale, which in turn affects the firm value and operating performance [29]. In this paper, the firm size is the natural logarithm of the total assets of the firm. It is calculated as follows:

$$Size_{it} = Ln\left(TOTAsset_{it}\right) \tag{10}$$

Where TOTAsset is the total assets of firm i in year t.

3. *Corporate growth rate.* Firm value is composed of current asset value and future growth opportunities. The faster a company grows, the higher its business performance (Zhou, Liu & Luo, 2022) [30]. In this study, the growth rate of a company is measured by (current year's net sales revenue—previous year's net sales revenue)/previous year's sales revenue.

4. *Research expenses and advertising expenses.* Research expense directly determines the update and iteration of the enterprise's products. With the investment in research expenses, the products launched by the enterprise will be more technological and in line with the market expectations. In addition, with the increase in Advertising Expenses investment, the products launched by the enterprise will appear in the public more frequently, which significantly improves the performance of the enterprise. R&D is the total amount of research and development expenditures incurred in the current period, including R&D expenditures charged to profit and loss and capitalized R&D expenditures (Pisano, 1990) [31]. SE is the cost of marketing and distributing the products of the enterprise including advertising, sales commissions, salaries, sales office expenses, and shipping costs (Venieris, Naoum & Vlismas, 2015) [32].

5. *Proportion of independent directors versus institutional ownership.* The independent director ratio is calculated as the percentage of independent directors to the total number of board members (Nguyen, 2010) [33]. The institutional shareholding ratio is quantified as the percentage of freely traded shares held by the institution to the number of outstanding shares (Boix, 2013) [34].

## 4. Data analysis

This chapter is based on the research sample in Chapter 3, which was selected as a sample of 113 Chinese Taiwanese companies and 113 Korean companies for the years 2017–2022, and was empirically analyzed using Stata 15.0.

### 4.1 Descriptive statistics

Firstly, descriptive statistics were conducted on the explained variables, explanatory variables, and control variables, and the statistical results are shown in Table 2 below:

As shown above, the average value of enterprise growth rate is maintained at around 14.0%, which means that the enterprise grows at a rate of 14.0% every year, but this indicator varies greatly from enterprise to enterprise, with the better-developed enterprises growing at a rate of 1,500%, and the enterprise realizing leaps and bounds, while the slower-developed enterprises even have a negative growth rate, and the enterprise shrinks by 84.9%. In the dimension of enterprise performance, the minimum value is 0.588 and the maximum value is 51.452, which is relatively more concentrated. In the enterprise size dimension, the minimum value is 3.696 and the maximum value is 12.790. Although the standard deviation is small, only 2.472, the data of the enterprise size is more unconcentrated and there is a large discrepancy because the enterprise size has been taken as a logarithm in the calculation. In terms of ESG scores, the minimum value is 21.2, the maximum value is 76.6, and the mean value is 51.8, with the data showing a normal distribution. In terms of ROE, the minimum value is -154.2 and the maximum value is 125.9 again showing great variability. In terms of ROA, the maximum value is 77.9 and the minimum value is -23.9, which like ROE shows poor variability. In terms of independent directors, the mean value is 42%, which matches the basic situation in Taiwan and Korea. In terms of institutional shareholding, some companies have 0% institutional shareholding, while some have as high as 97.8%, and the author has counted the companies with shareholding, and the top ten companies with institutional shareholding ratio Korea alone occupy nine seats. In terms of R&D investment and advertising investment, due to the differences in the industry and the nature of the enterprise, some enterprises do not need to make

**Table 2. Descriptive statistics.**

|  | Min | Max | Mean | SD |
|---|---|---|---|---|
| GROWTH | -84.884 | 1520.244 | 14.041 | 59.142 |
| Tobin's q | .588 | 51.452 | 1.972 | 2.472 |
| size | 3.696 | 12.790 | 8.133 | 1.506 |
| SCORE | 21.205 | 76.642 | 51.793 | 12.932 |
| ROE | -154.217 | 125.903 | 11.688 | 15.455 |
| ROA | -23.876 | 77.856 | 6.270 | 7.987 |
| Debt | .000 | 5.224 | .877 | .696 |
| Equity | 4.151 | 30.569 | 13.048 | 4.063 |
| Liquidity | .060 | 58.150 | 2.244 | 3.532 |
| Debt ratio | .000 | 85.373 | 22.160 | 15.414 |
| Sales | .001 | 3.152 | .834 | .497 |
| Director | 11.111 | 83.333 | 42.875 | 14.336 |
| NPS | -6008.736 | 265.451 | -.480 | 184.613 |
| Institutional | .000 | 97.807 | 33.516 | 15.639 |
| R&D | .000 | 24929171.000 | 269709.762 | 1690798.197 |
| advert | .000 | 13223600.000 | 177720.504 | 913757.339 |

R&D investment and advertising investment, so the minimum value is 0, while the average value of R&D is 269709.8, and the average value of advertising investment is 177720.5, i.e., most of the enterprises pay much more attention to R&D investment than advertising investment.

## 4.2 Selection of the model

After the introduction of the models in Chapter 3, the appropriate model was selected as the empirical model. The definition of the intercept term is shown as follows: (1) Ordinary Least Square method applies when all samples have the same intercept term; (2) Fixed effects model applies when the cross-sectional samples are allowed to have different intercepts; and (3) Random effects model assumes that the intercepts of the samples are all random variables.

Therefore the original hypothesis is rejected and the fixed effect model should be adopted.

## 4.3 Analysis of empirical results

**4.3.1 Analysis of overall regression results.** In this study, the Panel Data model was used for empirical analysis, and the fixed effect model was selected for regression analysis after the validation, and the results are shown in Tables 3 and 4:

As shown in the table above, in the fixed-effects model, the average ESG performance score of the regression results has a significant positive impact on Tobin's q enterprise business performance, with an impact coefficient of 0.007 (P = 0.015), which is consistent with the expectations of this paper, and hypothesis 1 is verified. Enterprises should incorporate the macro planning of ESG into the daily operation and management of the enterprise, and through the optimization and adjustment of the indicators, evaluate and integrate ESG into the daily management of the enterprise, to improve the enterprise business performance. In terms of control variables, ROE is positively correlated with business performance, and Hypothesis 2 is verified; however, enterprise size is negatively correlated with business performance, and Hypothesis 3 is rejected; the enterprise growth rate is not correlated with business performance, and Hypothesis 4 is rejected. The impact coefficient of R&D investment is 0.049 (P = 0.000), and the impact coefficient of advertising investment is -0.769 (P = 0.008), hypothesis 5 gets rejected, but R&D investment has a more significant impact compared to advertising investment, so hypothesis 5–1 is verified. The influence coefficient of the proportion of independent directors is -.0001 (P = 0.859), and the influence coefficient of institutional shareholding is .016 (P = 0.000), hypothesis 6 is rejected, but the influence of institutional shareholding on the business performance of the enterprise is greater than that of the proportion of independent directors has a more significant influence on the business performance of the enterprise, so hypothesis 6–1 is verified.

**4.3.2 Regression comparison analysis of Taiwanese and Korean firms.** To further explore what kind of impact ESG has on Taiwanese and Korean firms, and what the mechanism and degree of impact of ESG on the performance of Korean and Taiwanese firms, this section divides Taiwanese and Korean firms into different groups and conducts regression

**Table 3. Model summary.**

| Source | Ss | Df | MS | |
|---|---|---|---|---|
| Model | 5779.618 | 15 | 385.307 | Number of obs = 834 |
| Residual | 7691.890 | 823 | 9.346 | F(4,289) = 55.00<br>Prob>F = 0.000<br>R-squared = 0.649 |
| Total | 13471.508 | 838 | 16.075 | Root MSE = 2.293 |

**Table 4. Model regression results.**

| Tobin's q | Coef. | Std.err | t | P>|t| | 95% Conf. Interval | |
|---|---|---|---|---|---|---|
| Score | .007 | 0.007 | 2.43 | 0.015 | .003 | .031 |
| size | -.599 | .0604 | -9.93 | 0.000 | -0.719 | -.481 |
| Growth | .001 | .0012 | 1.20 | 0.229 | -.001 | .004 |
| ROE | .045 | .0055 | 8.26 | 0.000 | .035 | .056 |
| ROA | .066 | .009 | 7.38 | 0.000 | .048 | .083 |
| Debt | -.188 | .095 | -1.97 | 0.049 | -.376 | -.001 |
| Equity | .050 | .016 | 3.14 | 0.002 | .004 | .022 |
| Liquidity | .103 | .021 | 4.86 | 0.000 | .061 | 0.145 |
| Debt ratio | -.005 | .004 | -1.13 | 0.257 | -.014 | .003 |
| Sales | -.405 | .126 | -3.21 | 0.001 | -.652 | -.157 |
| Director | -.001 | .004 | -0.180 | 0.859 | -.010 | .008 |
| NPS | .001 | .000 | 1.48 | 0.140 | -.000 | .001 |
| Institutional | .016 | .004 | 3.76 | 0.000 | .007 | .024 |
| R&D | .049 | .006 | 7.65 | 0.000 | .036 | .061 |
| Advert | -.769 | -.290 | -2.66 | 0.008 | -.633 | -.221 |
| _cons | .179 | .362 | 0.500 | 0.621 | -.531 | .890 |

analyses. The models in Tables 5 and 6 are as follows:

$$
\begin{aligned}
\text{Tobin'Q}_{i,t} = {} & \propto_1 \text{Score}_{i,t} + \propto_2 \text{Size}_{i,t} + \propto_3 \text{ROA}_{i,t} + \propto_4 \text{R\&D}_{i,t} + \propto_5 \text{Director}_{i,t} \\
& + \propto_6 \text{Institutional}_{i,t} + \propto_7 \text{Advert}_{i,t} + \propto_8 \text{Debt}_{i,t} + \propto_9 \text{Equity}_{i,t} + \propto_{10} \text{Liquidity}_{i,t} \\
& + \propto_{11} \text{Dbet ratio}_{i,t} + \propto_{12} \text{Sales}_{i,t} + \propto_{13} \text{NPS}_{i,t} \quad\quad (11)
\end{aligned}
$$

Table 5 shows the data of South Korean enterprises and Table 6 shows the data of Taiwan enterprises. In the regression process of this paper, the index ROA was not included in the regression model. The reason is that the gap between the ROA of Taiwan enterprises and Korean enterprises is too large, and the technical level is obviously different, so the regression

**Table 5. Regression results of Korean firms' performance and ESG.**

| Tobin's q | Coef. | Std.err | t | P |
|---|---|---|---|---|
| Score | .017 | .007 | 3.022 | .000 |
| growth | -.789 | .083 | -9.558 | .000 |
| size | .004 | .002 | 1.642 | .139 |
| ROE | .005 | .005 | 0.992 | .322 |
| R&D | .010 | .012 | 2.863 | .003 |
| Director | .016 | .007 | 2.083 | .038 |
| Institutional | .437 | .007 | 6.789 | .000 |
| Advert | -.153 | -.321 | -4.761 | 0.000 |
| Debt | -.127 | .091 | -3.511 | .000 |
| Equity | .066 | .019 | 1.785 | .075 |
| Liquidity | .018 | .032 | .459 | .646 |
| Debt ratio | -.101 | .006 | 2.374 | .018 |
| Sales | -.184 | .165 | -5.217 | .000 |
| NPS | .057 | .003 | 1.326 | .182 |
| _cons | 8.076 | .516 | 15.653 | .000 |

**Table 6. Regression results of corporate performance and ESG in Taiwan, China.**

| Tobin's q | Coef. | Std.err | t | P |
|---|---|---|---|---|
| Score | .0131 | .009 | 2.373 | 0.012 |
| growth | -.656 | .086 | -7.638 | 0.000 |
| size | .002 | .001 | 1.843 | 0.067 |
| ROE | .024 | .006 | 8.040 | 0.000 |
| R&D | .063 | .008 | 8.048 | 0.000 |
| Director | -.010 | .010 | -1.015 | 0.312 |
| Institutional | .0125 | .004 | 0.207 | 0.025 |
| Advert | -.359 | -.181 | -1.993 | 0.147 |
| Debt | -.039 | .237 | -1.324 | .186 |
| Equity | .006 | .022 | .204 | .838 |
| Liquidity | .208 | .029 | 4.754 | .000 |
| Debt ratio | .232 | .007 | 6.041 | .000 |
| Sales | .050 | .164 | 1.570 | .117 |
| NPS | .043 | .000 | 1.156 | .248 |
| _cons | 4.3827 | .7487 | 5.852 | 0.000 |

results are different (Dionisio, Marcelo, et al.,2023) [35]. These factors include market structure, industry characteristics, degree of competition, management decisions, resource allocation, innovation ability, financial market tension, capital cost, debt ratio, economic cycle fluctuations, etc. They interact and jointly determine the ability of enterprises to use assets to create profits, resulting in differences in ROA.

The results are shown in Tables 5 and 6 below:

As shown in the above two tables, the ESG is both for the performance of Taiwanese firms and for the performance of Korean firms, but the degree of impact is different, ESG for Korean firms is more significant, but the degree of impact is smaller (p = 0.000, impact coefficient of 0.007), but for Taiwanese firms the impact is less significant but the impact coefficient is larger (p = 0.012, impact coefficient is 0.013). In the firm size dimension, for both Taiwanese and Korean firms, as firm size increases, performance improvement becomes more difficult, and the impact between firm size and performance is negative (p = 0.000, coefficient -0.789 for Korean firms; p = 0.000, coefficient -0.656 for Taiwanese firms). In the firm growth dimension, Korean firms' growth has a more significant effect on firm performance with an effect level of 0.002, while Taiwanese firms have no effect in this dimension. In the ROE dimension, the effect of gearing on firm performance is significant for Taiwanese firms, with an impact coefficient of 0.024, while Korean firms have no effect in this dimension. In the R&D investment dimension, R&D investment has a positive impact on the marketing performance of both Korean and Taiwanese firms; in the advertising investment dimension, Korean firms instead experience a downward trend in their business performance as their advertising investment increases, with an impact coefficient of -0.153, but advertising investment has no significant impact on the performance of Taiwanese firms. The proportion of independent directors has a non-significant effect on Taiwanese firms and a positive effect on Korean firms with complexity. In the dimension of institutional shareholding, its impact is positive for both Korean and Taiwanese firms, but the degree of change in the performance of Korean firms is much greater than that of Taiwanese firms as the proportion of institutional shareholding rises. To address the above phenomenon, the paper provides further discussion in the conclusion section.

**4.3.3 The impact of each ESG dimension for business performance.** In order to further explore the impact of the various dimensions of ESG business performance for business

performance, this paper splits the ESG composite score, and conducts regression analysis with Environmental, Social, and Governance as the explanatory variables respectively. The model is shown below:

$$\begin{aligned}
\text{Tobin}\prime Q_{i,t} = {}& \propto_1 E_{i,t} + \propto_2 S_{i,t} + \propto_3 G_{i,t} + \propto_4 \text{Size}_{i,t} + \propto_5 \text{ROA}_{i,t} + \propto_6 R\&D_{i,t} + \propto_7 \text{Director}_{i,t} \\
& + \propto_8 \text{Institutional}_{i,t} + \propto_9 \text{Advert}_{i,t} + \propto_{10} \text{Debt}_{i,t} + \propto_{11} \text{Equity}_{i,t} + \propto_{12} \text{Liquidity}_{i,t} \\
& + \propto_{13} \text{Dbet ratio}_{i,t} + \propto_{14} \text{Sales}_{i,t} + \propto_{15} \text{NPS}_{i,t}
\end{aligned} \tag{12}$$

The results are shown in Table 7 below:

The empirical analysis shows that ESG contributes to the improvement of firm performance in both Taiwanese and Korean firms. However, which dimension plays a major role is further discussed in this section. From the above empirical analysis, it can be seen that in the environmental dimension, its impact coefficient is 0.098 (P = 0.004), which is significant; in the social dimension, its impact coefficient is 0.075 (P = 0.009), which is significant; however, in the governance dimension, its impact coefficient is negative (P = 0.291), which does not affect the enhancement of corporate performance. Analyzing the reasons for this this paper suggests that in the governance dimension, enterprises need to meet the information disclosure requirements stipulated by local laws, often only meet the minimum needs of information disclosure when making disclosures, the implementation is insufficient, and there are serious surface governance problems, which leads to governance is not related to performance. In the environmental dimension, the current global advocacy of green finance and green production emphasizes the importance of environmental protection for enterprise development, and as the enterprise's score on the environmental dimension improves, its efficiency in the use of renewable resources also improves, which significantly narrows the enterprise's operating costs and improves enterprise performance. In the social dimension, an increase in a company's score on the social dimension reflects, to some extent, the way the company treats its employees, suppliers, customers, community, and society, which all reflect the company's corporate culture and inclusiveness. More and more consumers are paying attention to corporate

**Table 7. Regression analysis results of ESG Sub-dimensions.**

| Tobin's q | Coef. | Std.err | t | P |
|---|---|---|---|---|
| E | .098 | .004 | 2.859 | .004 |
| S | .075 | .005 | 2.462 | .009 |
| G | -.036 | .008 | -1.056 | .291 |
| growth | -.006 | .001 | -.258 | .796 |
| size | .331 | .057 | 9.586 | .000 |
| ROE | .191 | .008 | 3.823 | .000 |
| R&D | -.216 | .006 | -8.061 | .000 |
| Director | .103 | .005 | 3.383 | .001 |
| Institutional | .089 | .004 | 3.490 | .000 |
| Advert | .083 | .000 | 3.003 | .003 |
| Debt | -.073 | .090 | -2.878 | .004 |
| Equity | .062 | .015 | 2.451 | .014 |
| Liquidity | .086 | .022 | 2.785 | .005 |
| Debt ratio | .061 | .005 | 2.120 | .034 |
| Sales | -.056 | .121 | -2.285 | .022 |
| NPS | .051 | .000 | 1.907 | .057 |
| _cons | 4.436 | 0.581 | 7.635 | 0.000 |

social responsibility and sustainability practices. They may be more inclined to buy products and services from companies that are produced in an environmentally friendly and socially responsible manner. This trend has prompted companies to pay more attention to their ESG performance, as they realize that it will directly affect consumers' purchasing decisions. However, in the dimension of corporate governance, the management structure, decision-making process, transparency, code of ethics, and legal compliance of enterprises are all disclosed through their annual reports, which are usually characterized by "reporting the good news but not the bad news", and some of the negative impacts are only disclosed when mandated by the government, which results in an information asymmetry between the public and the enterprises, and the long run, a lack of information. This leads to information asymmetry between the public and enterprises, and in the long run, the public and the market no longer have positive attitudes toward enterprises, thus weakening the impact of corporate governance on business performance to a certain extent. It is worth noting that in terms of control variables, there is a significant difference between advertising investment and the previous regression results. The influence of advertising investment and Tobin Q value of Korean enterprises and Taiwan enterprises is negative (-.153 and -.359, respectively), but in the fractal regression, the influence of advertising investment and Tobin Q value is positive (.083). This paper believes that the reason for the coefficient instability is the variance discontinuities of the sample data collected in this paper (Agrrawal and Clark, 2009) [36].

## 5. Conclusion

By analyzing in-depth the ESG practices of listed companies in South Korea and Taiwan and their impact on business performance, this study found that ESG has had different impacts on companies in the two regions. Korean firms are more significantly affected by the ESG evaluation system, but their influence is relatively small, while Taiwanese firms are more affected but with weaker significance. This suggests that the effect of ESG implementation is influenced by region-specific factors, including government policies, market maturity, and firm-government relations.

Korean firms need to respond positively to the ESG rating system after its implementation due to strong government support and market demand. This resulted in a significant but weak impact of ESG scores on firm performance. On the contrary, Taiwanese firms are immature due to the overall level of development and the fact that some firms have a close relationship with the government, while others are more market-oriented, and this differentiation leads to a large but insignificant impact of the ESG evaluation system on Taiwanese firms.

Specifically in terms of firm size, according to data disclosed by Bloomberg, there is a negative correlation between firm size and business performance, both in South Korea and Taiwan. The implementation of the ESG evaluation system has forced firms to make organizational adjustments, and larger firms find it more difficult to respond quickly to ESG changes due to their more complex organizational structures, so changes in ESG orientation may lead to negative impacts of ESG changes on them. Conversely, smaller firms may be more likely to integrate and benefit from ESG practices due to their flexibility and adaptability. However, it is important to note that regardless of the region, there is a risk that ESG's impact on performance will be costly as firms grow to a certain size and may be characterized by boundary effects. Therefore, companies and governments should take comprehensive measures to ensure that ESG practices play a positive role within the appropriate boundaries, while also avoiding over-investing resources and impacting business performance. Ongoing monitoring, flexible strategies, and active collaboration with stakeholders are all key tactics for successfully addressing ESG boundary effects.

As for business growth, Korean firms' growth has a significant impact on business performance with an impact coefficient of 0.0063, while Taiwanese firms do not have a significant impact in this dimension. This may be related to Korean firms' aggressive international diversified economic strategy and post-epidemic market opening policy, whereas Taiwanese firms are still more dependent on the Chinese mainland region or intra-local market, resulting in a smaller share of international business and therefore relatively less active in ESG practices. Thus the issues of Taiwan firms' dependence on specific markets, conservatism in the external economy, and the single market may be detrimental to the effectiveness of ESG implementation. These issues also highlight the need for more cooperation and policy interventions by the international community in promoting sustainable development and ESG practices to avoid the possible risks associated with single-market dependence.

In the dimension of R&D investment and advertising investment, these two inputs should have an inverted U-shaped relationship with business performance. Since ESG is a new thing and is still in the growth period in the whole life cycle, R&D investment will have a positive impact on the performance of enterprises, which is in the first half of the inverted U-shape, i.e., the intensity of R&D investment of enterprises is far from reaching the optimal level, so the current R&D investment is mainly contributing to the performance of the enterprises, and therefore, we should continue to increase the strength of R&D and further improve the innovation ability and innovation level of enterprises. However, in the dimension of advertising investment, since ESG does not essentially regulate advertising investment, and the companies selected in this paper are listed companies with more mature marketing systems, it would be counterproductive to increase advertising investment.

In addition, in terms of shareholders' equity, the sample firms in Taiwan, most of which belong to the real sector, have a significant impact of their gearing ratios on firm performance, with an impact coefficient of 0.556. In contrast, Korean firms, due to their diversified business structure, especially the aggressive expansion of their international financial business, have a non-significant impact on shareholders' equity on firm performance.

In summary, the implementation of ESG has had different impacts on Taiwanese and Korean firms. Although ESG may hurt firm performance in the short term, when implementing ESG strategies, firms need to comprehensively consider the market environment they are in, their economic structure, and their internal factors to achieve long-term, sustainable development, and competitive advantage. For companies, ESG is not only a way to improve management efficiency, but also a strategic opportunity to create value in different regions and market conditions.

This paper finds that the effectiveness of ESG implementation is not only influenced by internal factors within the firm but also significantly externally influenced by international region-specific factors. This analytical approach, which combines internal and external factors, is not only applicable to the Asia-Pacific region but also provides a more complex and varied perspective for governments and businesses in other regions of the world to observe. These findings not only contribute to a deeper understanding of the evolution of ESG practices across economies but also provide important theoretical and practical references for companies as they face the challenges of globalization. These insights help companies integrate and adjust their strategies more effectively, as well as help them formulate ESG strategies that are in line with local realities and respond rationally to international trends in the post-epidemic era. This ensures that corporate development strategies are harmonized with the needs of the international ESG market.

In summary, this study provides an in-depth examination of the performance of listed companies in South Korea and Taiwan in terms of environmental, social, and governance (ESG) practices and their impact on business performance. The results of the study show that there

are significant differences in the impact of ESG practices on firms in these two regions. In South Korea, strong support from government policies and market demand led to the effective implementation of the ESG evaluation system, which resulted in a positive but limited impact on performance as a result of firms' improvement in ESG scores. In Taiwan, on the other hand, due to the immaturity of the overall level of development and high dependence on specific markets, firms face greater challenges in ESG practices, and their impact on performance is larger but less significant. In addition, it was found that there is a negative correlation between firm size and business performance, both in Korea and Taiwan. Large firms face difficulties in responding quickly to ESG changes due to the complexity of their organizational structure, while smaller firms may integrate ESG practices more effectively due to greater flexibility and adaptability. This finding emphasizes the need for a comprehensive approach when implementing ESG strategies to ensure that ESG practices work within the appropriate scope while avoiding over-investment of resources.

For Taiwan, it is recommended that firms should work on market diversification, reduce dependence on a single market, and explore more internationalization opportunities. The Taiwanese government should provide more support to promote firms' competitiveness in international markets, especially by encouraging more aggressive internationalization and market diversification strategies. As for South Korea, it is recommended that firms should strengthen their ability to innovate on their own and adapt to the international market while maintaining good cooperation with the government.

Finally, for investors focusing on ESG investments in Asia, this study provides important insights. Investors should emphasize how companies adapt to ESG requirements in their particular market environment and focus on companies' ESG performance as a key indicator for assessing their long-term sustainability and risk management capabilities.

Future research should analyze in greater depth the direct effect of ESG scores back on the relationship between firms' internal management mechanisms, external market changes, and the subtle effect these factors have on changes in ESG scores. In addition, the study should explore the aspects of firms' internal strategic adjustments and external market policy changes, which would provide more complex and diverse perspectives for governments and firms in other regions in the face of globalization challenges and promote the long-term sustainable development of firms.

The limitation of this study is that it fails to treat individual companies by industry and fails to take into account industry influences. Future research should provide an in-depth understanding of the link between corporate performance and ESG scores across industries through more detailed industry splits, and more clearly distinguish the relationship between internal management mechanisms and external market changes. At the same time, the impact of ESG practices on other key performance indicators of companies, such as profitability, operational efficiency, liquidity, financial structure, value-added, and enterprise value, should be considered to more comprehensively assess the practical significance of ESG practices on business management.

## Supporting information

**S1 Data. The data file name for Taiwan and Korea.**
(XLSX)

**S2 Data. The data file name for Korean.**
(XLSX)

**S3 Data. The data file name for Taiwan.**
(XLSX)

## Author Contributions

**Conceptualization:** Shaojie Fan.

**Data curation:** Shaojie Fan.

**Formal analysis:** Shaojie Fan.

**Investigation:** Shaojie Fan.

**Methodology:** Shaojie Fan.

**Project administration:** Shaojie Fan.

**Resources:** Shaojie Fan.

**Writing – original draft:** Shaojie Fan.

**Writing – review & editing:** Shaojie Fan.

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
