## [Decision Letter · Decision Letter 0]

11 Jul 2024

PONE-D-24-22051An Exploratory Study on the Impact of ESG on Business Performance - Focusing on Listed Companies in Korea and TaiwanPLOS ONE

Dear Dr. FAN,

Thank you for submitting your manuscript to PLOS ONE. After careful consideration, we feel that it has merit but does not fully meet PLOS ONE’s publication criteria as it currently stands. Therefore, we invite you to submit a revised version of the manuscript that addresses the points raised during the review process. Please submit your revised manuscript by Aug 25 2024 11:59PM. If you will need more time than this to complete your revisions, please reply to this message or contact the journal office at plosone@plos.org. Please include the following items when submitting your revised manuscript:A rebuttal letter that responds to each point raised by the academic editor and reviewer(s). You should upload this letter as a separate file labeled 'Response to Reviewers'.A marked-up copy of your manuscript that highlights changes made to the original version. You should upload this as a separate file labeled 'Revised Manuscript with Track Changes'.An unmarked version of your revised paper without tracked changes. You should upload this as a separate file labeled 'Manuscript'.

We look forward to receiving your revised manuscript.

Kind regards,

Marcelo Dionisio

Academic Editor

PLOS ONE

3. We are unable to open your Supporting Information file [plos_latex_template.aux, plos_latex_template.bbl, plos_latex_template.blg, and plos_latex_template.log]. Please kindly revise as necessary and re-upload.

Reviewers' comments:

Reviewer's Responses to Questions

**Comments to the Author**

1. Is the manuscript technically sound, and do the data support the conclusions?

Reviewer #1: Yes

Reviewer #2: Yes

2. Has the statistical analysis been performed appropriately and rigorously? 

Reviewer #1: I Don't Know

Reviewer #2: No

3. Have the authors made all data underlying the findings in their manuscript fully available?

Reviewer #1: Yes

Reviewer #2: Yes

4. Is the manuscript presented in an intelligible fashion and written in standard English?

Reviewer #1: Yes

Reviewer #2: Yes

5. Review Comments to the Author

Reviewer #1: It was very interesting reading this paper. Thank you for including me as reviewer. The paper provides a comprehensive analysis of ESG practices and their impact on business performance in Korea and Taiwan. The comparative approach between these two regions, considering their unique economic structures, is particularly insightful and adds significant value to the existing literature on ESG. As I am from Europe, it was very interesting for me to read it but also to know a lot of new things from this part of world. The use of empirical data from the Bloomberg database and the application of Tobin's q ratio as a measure of business performance are commendable methodological choices by my opinion. Overall, this paper makes a significant contribution to the field of ESG and business performance. With some minor enhancements, it has the potential to offer even deeper insights and broader implications for academia and industry practitioners. I encourage researchers to continue in this direction.

Reviewer #2: A well written and thought out paper on non-overlapping vector of ESG factors for Taiwan and Korea. Kindly look at the following few numerical items. Maybe I'm missing a point but perhaps elaborate on these items briefly or address them.

1. In Table 2 (Descriptive statistics), the minimum value for Debt ratio is 0.000. What percentage of your sample is all-equity? (Some of your metrics have 3 decimal places, others have 2 or 1, please make them uniform, in the US 2 is standard).

2. In Table 4 (Model Regression Results), the 95% confidence interval for the Equity variable is incorrectly reported as "-3.758 -0.001". This doesn't match the positive coefficient (0.0503) and p-value (0.002) reported for this variable. Please address this. It's likely a typo. (Some of your metrics have 3 decimal places, others have 4, please make them uniform, in the US 2 is standard).

3. In Tables 5 and 6 (regression results for Korean and Taiwanese firms), different sets of variables are included without explanation. For example, ROA is included in Table 4 but excluded in T5,6. Given that ROE varies widely for Taiwanese firms vs Korean firms, perhaps it could be included (or a brief reason provided, thank you). Also the Titles for T5 and T6 should be very similar, to convey it is a comparative panel table. Pls. standardize decimal length in these tables too. The technological footprint of Kr and Tw could be one of the factors for this ROE divergence. The significance of technology is well highlighted in Dionisio, Marcelo,et al. "Role of digital transformation in improving the efficacy ..."Journal of High Technology Management Research (2023).

4. In Table 7 (Regression Analysis Results of ESG Sub-dimensions), the coefficient for ROE is negative (-0.191), which seems to contradict the positive coefficient (0.045) reported for ROE in Table 4. Also a formal model using equation editor should be listed for T5,6,7 with Tobin's Q as the Dependent variable and others as the X variables.

5. In Table 7, the coefficient for Advertising (0.083) is positive, somewhat contradicting the negative coefficients reported for this variable in previous tables. Or please elaborate on it briefly. It is possible there is factor instability (which is fine), Agrrawal and Clark (2009), attribute factor/coefficient instability to variance discontinuities and note their impact on liquidity of assets "Multivariate Liquidity Score and Ranking Device for ETFs." Academy of Financial Services (2009).

A somewhat larger concern is the way you have narrowed down to 110 or so firms. From your Bloomberg screen [file is attached for reference] I can see that you eliminated companies without a full set of variables. This introduces Survivorship bias, as only the strong and mature companies will remain in existence starting 2017 to 2022. Please elucidate on how you controlled for this issue.

I found this point to be well brought out in the paper: In Taiwan, ESG practices, while emphasizing local culture and community integration, may lead to higher costs and less efficient practices. This response-oriented approach can result in inefficiencies compared to South Korea's more systematic strategy. In contrast to Taiwan, in South Korea, ESG practices are heavily reliant on government guidance and the promotion by large corporations, leading to insufficient participation of SMEs. This dependence reduces market autonomy and innovation drive. The subsequent performance and liquidity impact of these ESG variations can be further researched applying the (Agrrawal and Clark, 2009 process). Additionally Similar studies applying Sharia criterion have been applied to the Australian market [El Saleh and Jurdi. "Stock performance under alternative Shariah screening methods: Evidence from Australia." Accounting & Finance (2021)].

The paper can be streamlined nicely after addressing some of the points and incorporating feedback which will improve visibility as well. Good work, thank you.

6. PLOS authors have the option to publish the peer review history of their article (what does this mean?). If published, this will include your full peer review and any attached files.

Reviewer #1: No

Reviewer #2: No

---

## [Author Response · Author response to Decision Letter 0]

19 Aug 2024

I sincerely thank the editor and al reviewers fortheir valuable feedback that Ihave used to improve the quality of our manuscript.The reviewer comments are laid out below in italicized font and specific concerns have been numbered.My response is given in normal font and changes/additions to the manuscript are given in the blue text.

Reviewer #1：

It was very interesting reading this paper. Thank you for including me as reviewer. The paper provides a comprehensive analysis of ESG practices and their impact on business performance in Korea and Taiwan. The comparative approach between these two regions, considering their unique economic structures, is particularly insightful and adds significant value to the existing literature on ESG. As I am from Europe, it was very interesting for me to read it but also to know a lot of new things from this part of world. The use of empirical data from the Bloomberg database and the application of Tobin's q ratio as a measure of business performance are commendable methodological choices by my opinion. Overall, this paper makes a significant contribution to the field of ESG and business performance. With some minor enhancements, it has the potential to offer even deeper insights and broader implications for academia and industry practitioners. I encourage researchers to continue in this direction.

I feel great thanks for your professional review work on our article. As you are concerned, there are several problems that need to be addressed.According to your nice suggestions, we have made extensive corrections to our previous draft.The specific corrections are as follows:

1.I feel sorry for our carelessness.In my resubmitted manuscript,the typo is revised. Thanks for your correction.

2.I tried my best to improve the manuscript and made some changes to the manuscript. These changes will not influence the content and framework of the paper. 

3.We sincerely appreciate the valuable comments. Wehave checked the literature carefully and added more references on Digital transformation and Model stability in the revised manuscript.

4. Some data errors were corrected, and the instability of the variable of advertising input was explained and explained.

Reviewer #2：

1. In Table 2 (Descriptive statistics), the minimum value for Debt ratio is 0.000. What percentage of your sample is all-equity? (Some of your metrics have 3 decimal places, others have 2 or 1, please make them uniform, in the US 2 is standard).

The company with a debt ratio of 0 is STUDIO DRAGON CORP of South Korea, code 253450.

Unified decimals, the article unified into 3 decimals. The reason is that some data is too small, if it is two decimals, some data is 0.00, three decimals are more accurate.

2. In Table 4 (Model Regression Results), the 95% confidence interval for the Equity variable is incorrectly reported as "-3.758 -0.001". This doesn't match the positive coefficient (0.0503) and p-value (0.002) reported for this variable. Please address this. It's likely a typo. (Some of your metrics have 3 decimal places, others have 4, please make them uniform, in the US 2 is standard).

This problem does exist because of a typo, because Debt and Equity are adjacent lines, and later in the writing process, 95% of these two questions were incorrectly confconfed. Written for the same, now modified. Thank you for your careful review!

3. In Tables 5 and 6 (regression results for Korean and Taiwanese firms), different sets of variables are included without explanation. For example, ROA is included in Table 4 but excluded in T5,6. Given that ROE varies widely for Taiwanese firms vs Korean firms, perhaps it could be included (or a brief reason provided, thank you). Also the Titles for T5 and T6 should be very similar, to convey it is a comparative panel table. Pls. standardize decimal length in these tables too. The technological footprint of Kr and Tw could be one of the factors for this ROE divergence. The significance of technology is well highlighted in Dionisio, Marcelo,et al. "Role of digital transformation in improving the efficacy ..."Journal of High Technology Management Research (2023).

The regression data of some variables in Table 4 and Table 5 are supplemented to make the article more complete. However, ROA was not included in the regression model constructed in this paper. According to Dionisio, Marcelo,et al (2023), the paper explained why ROA was not included in the model, and carried out scalability analysis.

4. In Table 7 (Regression Analysis Results of ESG Sub-dimensions), the coefficient for ROE is negative (-0.191), which seems to contradict the positive coefficient (0.045) reported for ROE in Table 4. Also a formal model using equation editor should be listed for T5,6,7 with Tobin's Q as the Dependent variable and others as the X variables.

The models for Tables 5 and 6 are written out above Tables 5 and 6.

The model for Table 7 is written out above Table 7.

In the regression analysis of each subdimension, the regression results of growth, size, ROE and R&D were corrected. This is really the author's oversight, thank you for pointing out the shortcomings of this article, based on your comments, I believe the quality of the article will become better

5. In Table 7, the coefficient for Advertising (0.083) is positive, somewhat contradicting the negative coefficients reported for this variable in previous tables. Or please elaborate on it briefly. It is possible there is factor instability (which is fine), Agrrawal and Clark (2009), attribute factor/coefficient instability to variance discontinuities and note their impact on liquidity of assets "Multivariate Liquidity Score and Ranking Device for ETFs." Academy of Financial Services (2009).

The correlation between advertising investment and Tobin Q of Korean enterprises and Taiwan enterprises is negative, but in the regression of the fractal dimension, the results can be significantly different from each other. Therefore, this paper has listened to your ideas and combined agrawal and Clark(2009) to explain the reasons for the instability of the coefficient. This will improve the quality of the article.

I appreciate the reviewers' thorough evaluation and recognize the major concerns raised regarding the clarity of my main argument and the completeness of the theoretical framework.I sincerely appreciate the time and effort invested by the reviewers in evaluating our manuscript. I look forward to any additional feedback or suggestions.

Sincerely，

Corresponding author FAN SHAOJIE

---

## [Decision Letter · Decision Letter 1]

1 Sep 2024

An Exploratory Study on the Impact of ESG on Business Performance - Focusing on Listed Companies in Korea and Taiwan

PONE-D-24-22051R1

Dear Dr. FAN,

We’re pleased to inform you that your manuscript has been judged scientifically suitable for publication and will be formally accepted for publication once it meets all outstanding technical requirements.

Kind regards,

Marcelo Dionisio

Academic Editor

PLOS ONE

Additional Editor Comments (optional):

Reviewers' comments:

Reviewer's Responses to Questions

**Comments to the Author**

1. If the authors have adequately addressed your comments raised in a previous round of review and you feel that this manuscript is now acceptable for publication, you may indicate that here to bypass the “Comments to the Author” section, enter your conflict of interest statement in the “Confidential to Editor” section, and submit your "Accept" recommendation.

Reviewer #2: All comments have been addressed

2. Is the manuscript technically sound, and do the data support the conclusions?

Reviewer #2: Yes

3. Has the statistical analysis been performed appropriately and rigorously? 

Reviewer #2: Yes

4. Have the authors made all data underlying the findings in their manuscript fully available?

Reviewer #2: Yes

5. Is the manuscript presented in an intelligible fashion and written in standard English?

Reviewer #2: Yes

6. Review Comments to the Author

Reviewer #2: The authors have been very thorough and responsive to technical feedback. It is a very well produced paper.

7. PLOS authors have the option to publish the peer review history of their article (what does this mean?). If published, this will include your full peer review and any attached files.

Reviewer #2: No

---

## [Editor Report · Acceptance letter]

25 Sep 2024

PONE-D-24-22051R1 

PLOS ONE

Dear Dr. FAN, 

I'm pleased to inform you that your manuscript has been deemed suitable for publication in PLOS ONE. Congratulations! Your manuscript is now being handed over to our production team.

Kind regards, 

on behalf of

Professor Marcelo Dionisio 

Academic Editor

PLOS ONE